# Trawl Fishing Fleet Operations Used to Illustrate the Life Cycle of the Southern Brown Shrimp: Insights to Management and Sustainable Fisheries

Ualerson I. Peixoto [1,2,3,4,*], Adauto S. Mello-Filho [1], Bianca Bentes [1,2] and Victoria J. Isaac [1,2]

1   Programa de Pós-Graduação em Ecologia Aquática e Pesca, Núcleo de Ecologia Aquática e Pesca da Amazônia (NEAP), Universidade Federal do Pará (UFPA), Avenida Perimetral 2651, Belém 66077-530, Pará, Brazil; filho.adauto@gmail.com (A.S.M.-F.); bianca@ufpa.br (B.B.); visaac@ufpa.br (V.J.I.)
2   Laboratório de Biologia Pesqueira e Manejo dos Recursos Aquáticos, Grupo de Pesquisa em Ecologia, Manejo e Pesca na Amazônia (GEMPA), Avenida Perimetral 2651, Belém 66077-530, Pará, Brazil
3   Departamento de Oceanografia e Pescas, Universidade dos Açores, Rua Prof. Dr. Frederico Machado, 4, 9900-138 Horta, Portugal
4   Okeanos—UAc Instituto de Investigação em Ciências do Mar, Universidade dos Açores, Rua Prof. Dr. Frederico Machado, 4, 9900-138 Horta, Portugal
*   Correspondence: ualerson.ip.silva@uac.pt

**Abstract:** The purpose of this study was to better understand the life cycle of brown shrimp along the Amazon Continental Shelf by using spatial and temporal trawl fleet activities. A total of 208,121 specimens and 1281 trawls were studied throughout the course of 13 years of shrimp size composition. To investigate differences in length composition between fishing grounds, months, and depth, a PERMANOVA analysis was employed. A geographic information system was developed for environmental characterisation and spatiotemporal trawl fleet distribution. Our findings show that the industrial trawl shrimp fleet has a close relationship with shrimp biological characteristics, following shrimp migration patterns in different months, locations, and depths during different stages of their life cycle, and that this fleet acts on two-yearly cohorts. The management measures of limiting effort (number of vessels) appear enough to avoid overcapitalisation, but the closed period and a proposed no-take fishing zone appear insufficient for what was originally proposed. Ecosystem-based management strategies should be addressed immediately because they would be far more effective than traditional fishery-based management measures in promoting sustainable fishing.

**Keywords:** trawl shrimp fishery; shrimp size composition; no-take zone fishing; *Penaeus subtilis*; shrimp fishery management; Amazon coast





## 1. Introduction

Shrimps are important fishing commodities in the world's tropical and subtropical marine environments [1,2]. Every year, the trawl shrimp fishery harvests 3.4 million tons of shrimp, putting the worldwide seafood trade at around USD 10 billion [3,4]. Shrimp fisheries can be carried out on an industrial or artisanal scale, and the shrimp fleet can operate in coastal lagoons, estuaries, or offshore, where it is performed by trawlers [3,5–7].

On the Brazilian coast, four shrimp species are caught, and they represent at least 45% of the total crustaceans caught by the country: *Penaeus paulensis* (Perez-Farfante, 1967) (São Paulo shrimp), *Penaeus subtilis* (Perez-Farfante, 1967), (Southern brown shrimp), *Penaeus brasiliensis* (Latreille, 1817) (Red-spotted shrimp), and *Xiphopenaeus kroyeri* (Heller, 162) (Seabob shrimp) [8]. In addition, a recently described shrimp species, *Penaeus isabelae* [9], is not reported on landings, due to difficulties with taxonomic identification [10].

The Amazon Continental Shelf (ACS) in northern Brazil is home to one of the world's most important shrimp fishery areas, where the industrial trawl shrimp fleet works [11–14]. Bottom

trawls are used in the industrial trawl shrimp fleet, with two nets working simultaneously in one boat. A jib or flat net is a type of net that features two "wooden doors" to help keep the nets open. Long steel wires tow the nets on each side of the vessel. Trawls are undertaken on mud and sandy seabeds at depths ranging from 10 to 100 m, with a preference for 50 to 60 m, which mostly target Penaeidae southern brown shrimp, *P. subtilis* [6,13,15,16].

*Penaeus subtilis* is distributed throughout the Atlantic Ocean's tropical regions, from Cuba to Rio de Janeiro, Brazil [1,17]. It has a typical Penaeidae complex life cycle, with several larval stages and migration patterns [18]. Post-larval and juvenile Penaeidae species inhabit coastal waters in habitats such as estuaries, lagoons, mangroves, and bays; however, adults migrate to offshore areas in the ocean and spawn there when they reach maturity [18,19]. The eggs and larvae are subsequently carried to coastal waters, by the marine hydrodynamic process, where the juveniles can develop safely and with sufficient food sources [3,19,20]. After the juvenile stages, they migrate back to offshore waters for maturation and spawning [3,20]. Mortality rates are high after the spawning season [3,18]. The longevity of *P. subtilis* is between one and two years [12,21].

Shrimp fishing is the primary fishery export commodity in many developing nations, and it is a significant social and economic activity [3]. At ACS, the shrimp fishery is a significant industry that generates both direct and indirect employment opportunities, as well as stimulates regional and national economies [22]. In 2007, the year of the last official Brazilian fishery statistics, shrimps were the second most important fishery product in Brazil, with 17,217 tons caught, reaching a revenue of USD 79.9 million. At ACS, 2091 tons of southern brown shrimp were caught, representing revenues of USD 12 million [23,24]. From the early 1960s through the 1980s, this fishery produced increased catches, with the highest catches in 1987. Since then, despite considerable yield oscillations with a negative trend in this fishery, the exploitation level is still below the MSY [14,16].

The Brazilian government imposed several restrictions on shrimp fisheries as management measures. The restrictions include a two-month yearly closure season between December and February [25], as well as a limit of 101 fleet vessels [26] and the prohibition of trawling activities near the coast [25]. This fisheries management model has been used on the north shore since 1980 [25–27], even though its effectiveness in improving the fishery or protecting the shrimp population has yet to be proven. The Brazilian Environmental Institute (IBAMA) recently suggested a no-take zone for fishing in the "Lixeira" (trash can in Portuguese) zone, which is closer to the Amazon River mouth. The Lixeira is a region where small shrimps and other aquatic resources can be found in great quantities, indicating a recruitment area [28–30].

Studying the dynamics of the fishing fleet and relating this to the movements and size structure of the shrimp population through the coast is essential to understand the complex life cycle and recruitment dynamics [31,32], to identify the main areas of recruitment and habitat use of shrimp at ACS, to detect potential fishing no-take areas [33], and to discuss the adequacy of the closed season [7]. This information is needed to improve fisheries management plans and develop approaches toward economic and environmentally sustainable fishery [34]. The distribution of southern brown shrimp along South America's northern coast, from Venezuela's Orinoco River to Brazil's Parnaiba River, is well documented [6,11]. However, the population movement pattern and ACS distribution pattern are still unknown. This research aims to gain access to the spatial and temporal trawl fleet dynamics in order to better understand the brown shrimp life cycle along the Amazonian coast and explore management implications.

The regional and temporal distribution of fishing fleets, as well as their relationship to shrimp growth and life cycle, may provide new management insights. As a result, the following questions will be addressed in this paper: (i) Which region has the greatest amount of available recruits? (ii) When may the most significant numbers of recruits be detected? (iii) Can the fleet's spatial and temporal dynamics be explained as a response to shrimp migrations along the coast? (iv) Does *Penaeus subtilis* have a migration pattern that is similar to that of other Penaeidae species?

## 2. Materials and Methods

### 2.1. Study Area

The southern brown shrimp fishery is carried out by industrial trawl vessels on the ACS off northern Brazil. The fishing grounds extend from the Oiapoque River (border between French Guiana and Brazil) to the Parnaiba River (between the Piaui and Maranhão states, Brazil). The highest trawl fishing effort occurs off Pará and Amapá states [6,16] (Figure 1). The Amazon and Pará rivers release a significant amount of water into this region, creating a complex habitat with high biological productivity that supports a high diversity and biomass of commercial and ecologically important aquatic species. Due to these features, most of these locations are appropriate for demersal fishing [35,36].

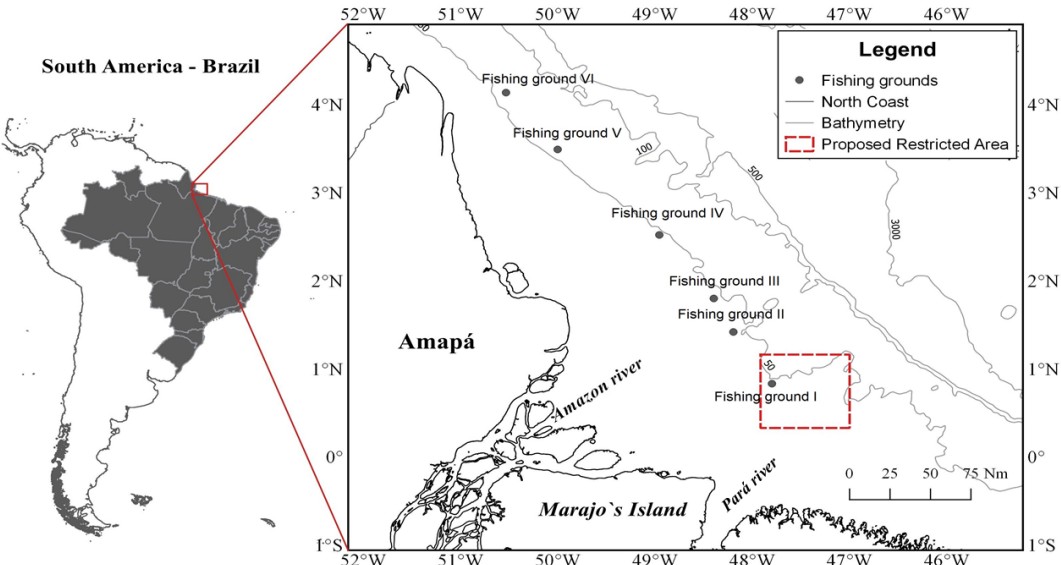

**Figure 1.** The Amazon Continental Shelf, northern Brazil, and the midpoint of the main fishing grounds.

The Amazon region experiences high-intensity rainfall during the first semester of the year, which is known as the rainy season, and low pluviometry during the second semester, which is known as the dry period. The Amazon River's discharge follows the rainfall cycle, with the highest volume in the first semester (average of 220,000 m$^3$ s$^{-1}$) and the lowest in the second semester (average of 100,000 m$^3$ s$^{-1}$) [37–40]. This significant discharge flows into the ocean, generating a surface freshwater plume that transports about 800 × 106 tonnes of suspended particles, as well as sediments, nutrients, and salinity variations [35,36,41].

This area is also influenced by the North Brazil Current, which flows from Northeast to Northwest, carrying nutrients and sediments along the Brazilian continental shelf [42]. The trade winds blow toward the northwest from December to May, and toward the Southeast from June to November [43]. All these fluvial, marine, and atmospheric processes combine to form a high-physical-energy environment with a seasonal dynamic that controls salinity, nutrients, sediments, and the dispersion of the Amazon freshwater plume in the ACS [43–45].

### 2.2. Shrimp Data

Individual southern brown shrimp samples were collected from vessels in the industrial trawl fishing fleet from 2000 to 2007, in 2010, and again from 2014 to 2017, for a total of 13 years. During these seven days, all trawls were documented, with information on the date, starting time, duration (h), initial and final latitude and longitude, depth (m), and fishing ground location. During the capture procedures, the samples were recorded onboard for seven days. A random sample of shrimps was obtained from each trawl, and their total length was recorded. Different months were recorded each year, leaving a gap in some months. As a result, the measurements were grouped by month, providing a length

dataset encompassing February to December. Because January falls during the closure period, no data were collected in that month.

### 2.3. Data Analysis

The overall length means of shrimps per trawl was used in all analyses. The spatial and temporal fleet distribution on the ACS was described using a Geographic Information System (GIS). We developed images with substrate and bathymetry for the environmental characterisation, and the trawls were plotted for each month of the year. The Brazilian Government already has a suggestion of a no-take area, from 00°20′ N to 1°10′ N and 47°00′ W to 47°55′ W [28]. As a result, the intended no-take area's limits were always marked on the maps. A kernel density analysis was used to estimate fishing intensity. The shapes used for this analysis were obtained from national geo-referenced database platforms (www.ibge.gov.br, (accessed on 22 May 2020)), http://www.cprm.gov.br (accessed on 30 June 2020). Qgis 3.0 (Open-Source Geospatial Foundation (OSGeo), Chicago, IL, USA) was used to conduct these analyses.

The main fishing grounds were identified and assigned numbers ranging from I to VI (adapted from [6]). To test for shrimp size differences between depths, the total depth of each trawl was used at 15 m class intervals. The proposed no-take zone is denoted by the red dashed line (Figure 1). The nonparametric multivariate permutational test PERMANOVA (Permutational Multivariate Analysis of Variance) was used to examine the difference in total mean length between fishing grounds, months, and depths, using Euclidian distance as a linkage function with 9999 permutations [46]. The fishing grounds, depth, and months were used as fixed factors. To compare the differences in total length means between the groups, a paired test was used.

According to [47], the criterion of 12.65 cm total length, which is the mean length at first maturation of southern brown shrimp taken on the ACS, was determined to classify shrimps as juveniles and adults. All statistical analyses were performed using "PERMANOVA" package 0.2.0 in R statistical software v.4.1.2 (R Core Team, Vienna, Austria).

## 3. Results

### 3.1. Spatial and Temporal Distribution of Total Length Mean

There were a total of 208,121 specimens measured. There were significant differences in overall mean length ($F = 228.4$; $p < 0.001$), months ($F = 58.5$; $p < 0.001$), and depths ($F = 6.6$; $p < 0.001$) between fishing grounds. From the Amazon estuary to the northwest, the mean length gradually increased. Fishing grounds I and II had the shortest mean lengths, with no significant differences. Fishing ground III had an intermediate length mean, whereas fishing grounds IV, V, and VI had the biggest sizes, with no difference. The mean lengths also vary by month, starting with the smallest values in October–November and increasing to the highest in September the following year. In March and September, there is more dispersion in the distribution of lengths around the mean lengths. Mean lengths differ between depths as well. The smallest shrimps were captured at depths of up to 60 m. Between 30 and 60 m, the average shrimp size was statistically smaller. Larger sizes were caught at a depth of 60 m (Table 1, Figure 2).

**Table 1.** Results of PERMANOVA testing differences in total length mean between fishing grounds, months, and depths. F.G. = Fishing grounds; *** = $p < 0.001$.

| Source | df | SS | MS | F. Model | $R^2$ | Significance |
|--------|-----|---------|---------|----------|---------|--------------|
| F.G. | 5 | 765.16 | 153.031 | 228.387 | 0.30177 | *** |
| Months | 10 | 391.99 | 39.199 | 58.501 | 0.15460 | *** |
| Depths | 4 | 17.64 | 4.411 | 6.583 | 0.00696 | *** |
| Residuals | 1124 | 753.14 | 0.670 | | 0.29214 | |
| Total | 1280 | 2535.58 | | | 1.00000 | |

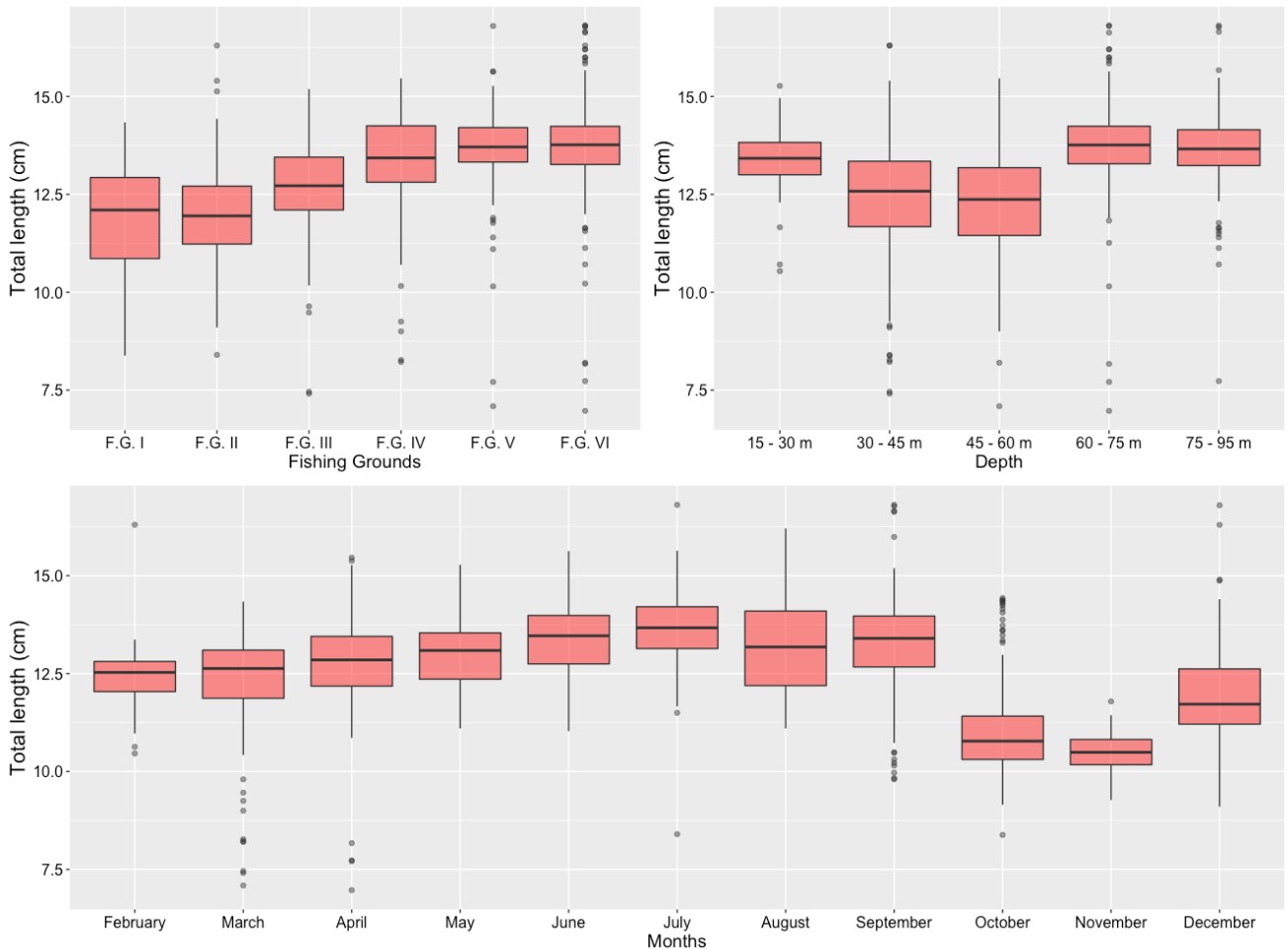

**Figure 2.** Distribution of average total length (cm) and standard deviation (95%) by fishing ground, depth, and months on the Amazon Continental Shelf. F.G. = Fishing Grounds.

### 3.2. Spatial and Temporal Fleet Dynamic

The activities of 1281 trawlers were examined, totalling 5127.50 h of fishing operations. In ACS, the total fishing area was assessed to be 112,699.7 km². Each trawl took an average of 4:30 h $\pm$ 1.39 (average and standard deviation). The majority of the trawls were conducted in fishing grounds I, II, and III (54.10%) and were centred in the waters off the coast of Pará State. Trawlers were observed all year in fishing grounds I and II, but they were more concentrated in October and November; then, the fleet's area of operation expanded farther to the northeast, to fishing grounds III and IV, between December and February. In March and April, however, the concentration was close to fishing grounds I and II once more. From May to September, operations in fishing grounds IV, V, and VI were concentrated (Figure 3). Almost all of the fishing operations took place around the 50 m isobath between the river mud seabed and the sandy substrate. The fleet fished in fishing areas I, II, and III at an average depth of 41.40 m $\pm$ 4.83, mostly on muddy and sandy seabeds. Trawlers operate on grounds connected with a sandy and sand and/or gravel branching coralline algae substrate in fishing grounds IV to VI from May to September, with an average depth of 68.64 m $\pm$ 15.57 (Figure 4).



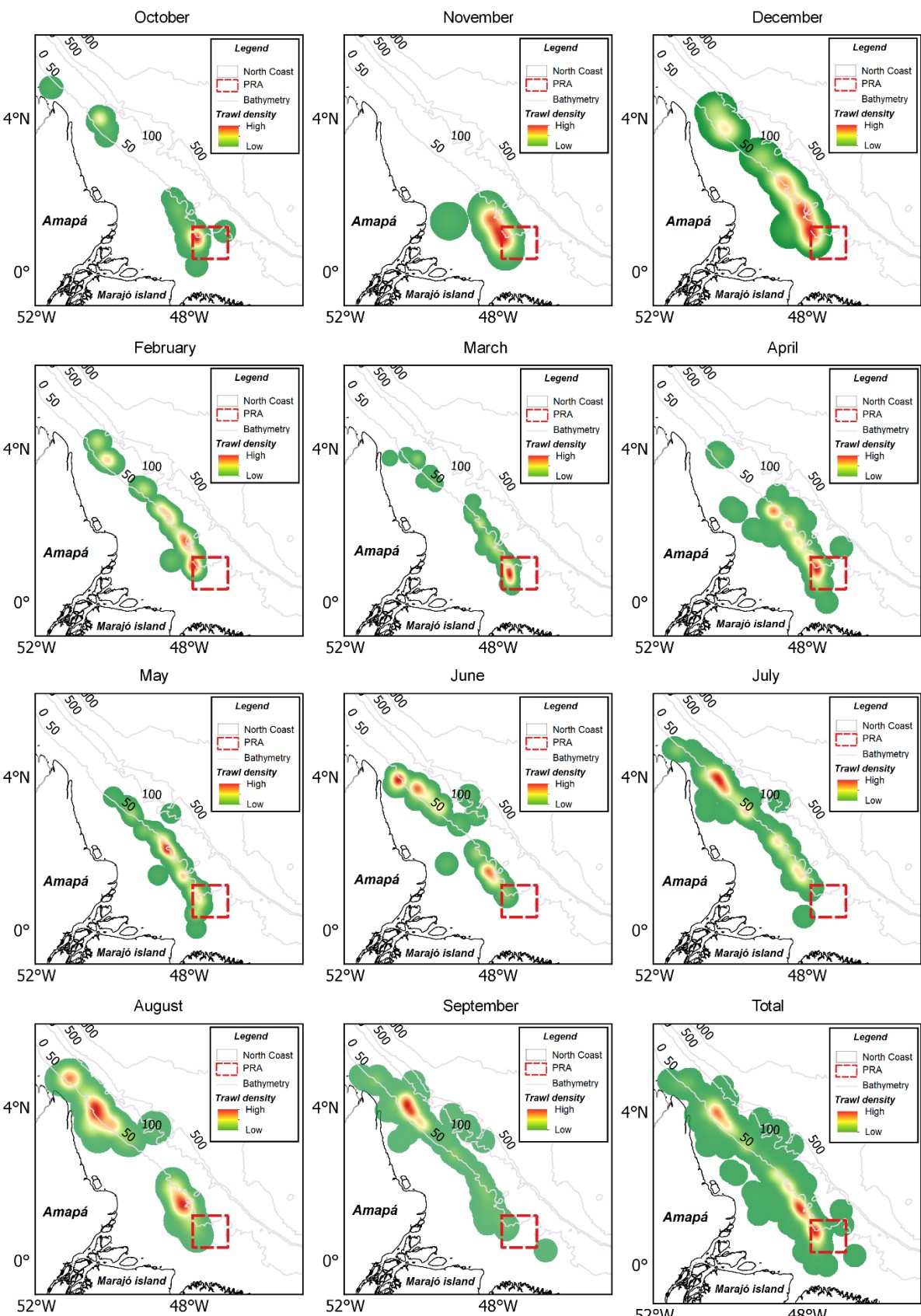

**Figure 3.** Spatial and temporal intensity and bathymetric structure (meters) distribution of shrimp trawlers on the amazon continental shelf. PRA: Proposed Restricted Area.

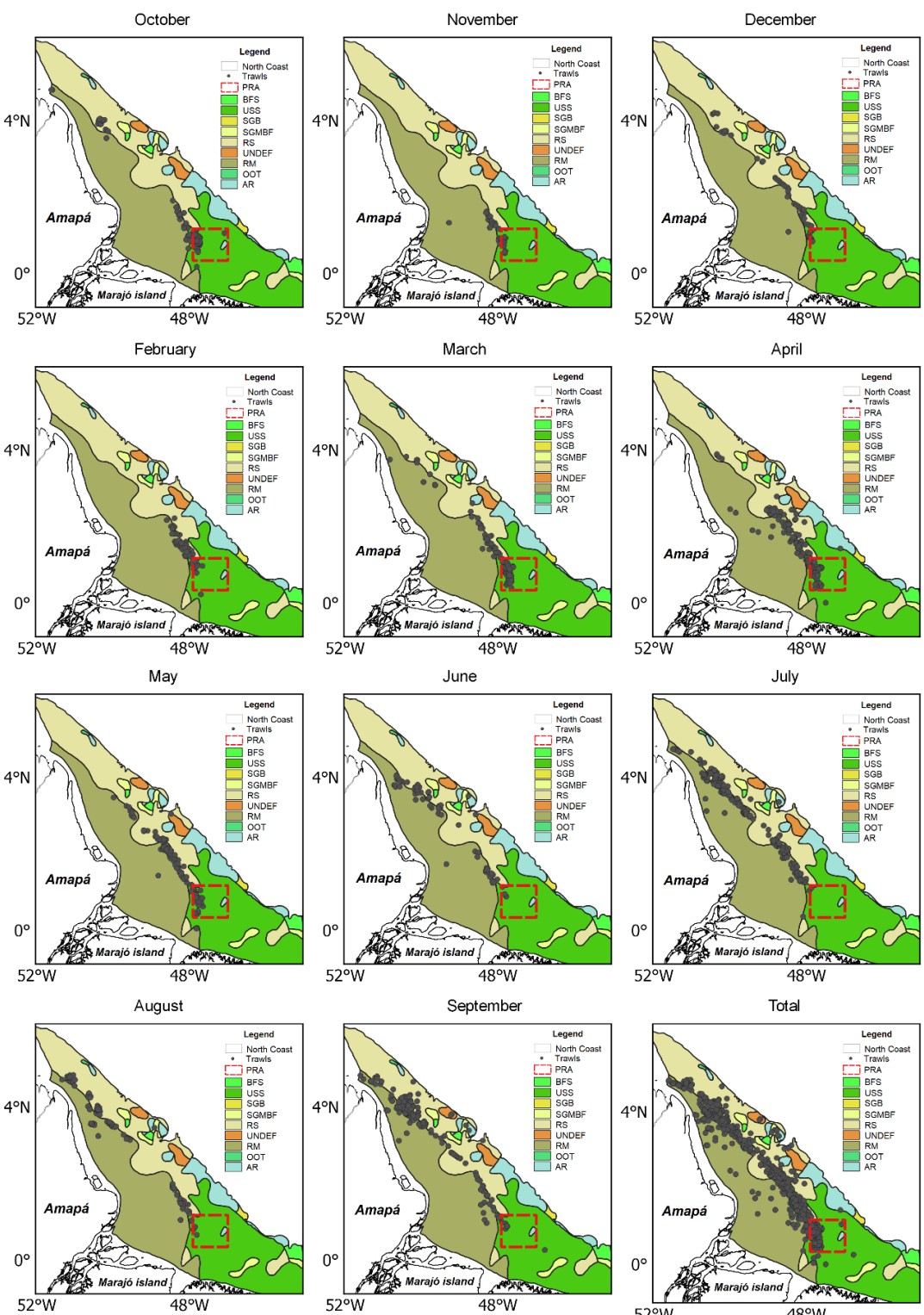

**Figure 4.** Spatial and temporal distributions of shrimp's trawlers by sedimentary structure on the Amazon continental shelf. PRA: Proposed restrict area; BFS: Benthic foraminifera sand; USS: Unspecified supplier sand; SGB: Sand and/or gravel of branching coralline algae; SGMBF: Sand and/or gravel of mollusks and benthic foraminifera; RS: River sand; UNDEF: Undefined; RM: River mud; OOT: Oolites; AR: Algae reef.

### 3.3. Spatial and Temporal Shrimp Distribution Pattern

The dynamics of the distribution of the trawling fleet between fishing grounds coupled with the pattern of monthly distributions of mean sizes allows the proposal of a spatiotemporal distribution pattern of *P. subtilis* along the year following a northwest route in the ACS. Juvenile individuals start moving from the estuaries into the ocean from October to December with approximately ~11.24 cm average total length. They are caught in fishing grounds I and II in waters up to 60 m in depth. Another pulse of recruitment seems to happen between March and April when the fleet also concentrates on fishing grounds I and II again. After these two peaks of recruitment, shrimps reach maturity, performing a migration to deeper waters. On fishing grounds, III and IV shrimps reach an average size of ~12.61 cm and can be found from February to April. From May up to September, the largest shrimps (average ~13.30 cm) are caught at fishing grounds V and VI (Figure 5).

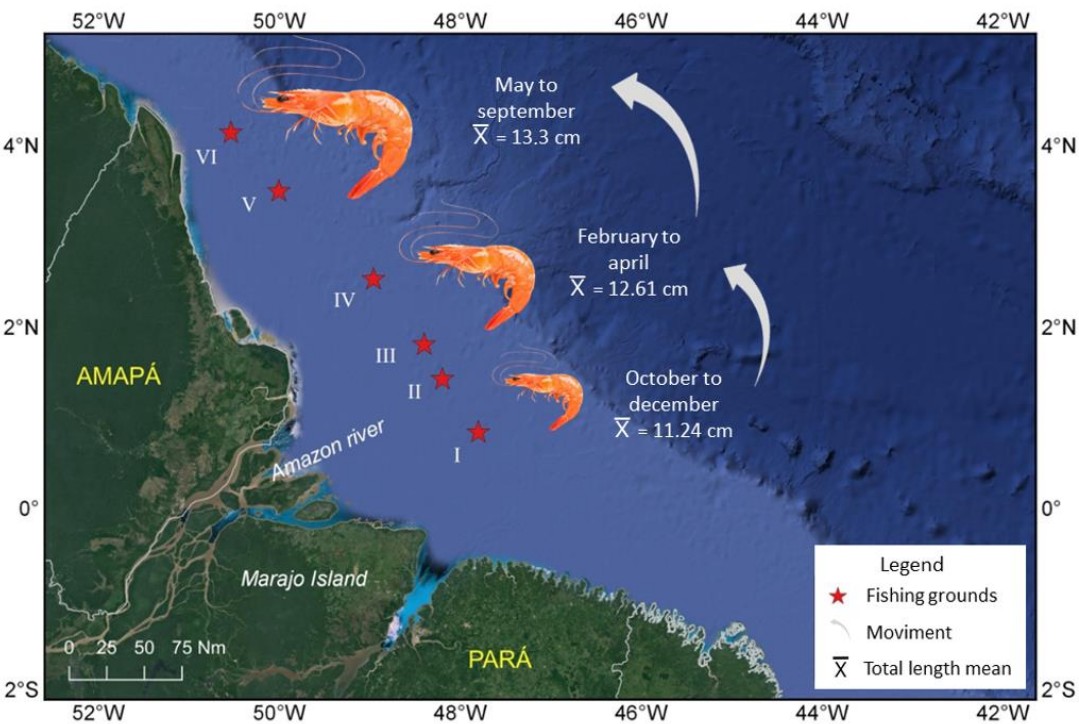

**Figure 5.** Diagram of the shrimp migration pattern on the Amazon Continental Shelf. FG = fishing ground; $\overline{X}$ = total length mean by area.

### 3.4. No-Take Area and Spatiotemporal Dynamics

Fishing operations do not occupy the full region specified by the government as a potential exclusion zone for a trawler (Figures 3 and 4). According to the data, the SE section of the proposed area is never utilised by the fleet at any time of year. The fleet uses only the NW corner of the circumscribed square as fishing grounds. The most intense use of this area is from October to December and from March to May, with peaks in November and April. The months when the fleet is on the proposed exclusion zone correspond to the months when shrimps are smaller, between 11 and 12 cm in total length, i.e., recruits and young individuals are being caught in this region.

## 4. Discussion and Conclusions

### 4.1. Spatiotemporal Distribution of Fishing Fleet and Size Composition

The shrimp fleet's temporal and spatial patterns, which follow the most abundant resource areas, help us explain the distribution pattern of southern brown shrimp. Even though the shrimp population's abundance varies widely, fishermen retain specific knowledge of fishing resources that is critical for optimising yield, economic return, and effort

by identifying places where individuals are more concentrated and fishing operations are more suitable [48,49]. Fishers provide a wealth of precise information for understanding fisheries dynamics and environmental marine resources [50]. The accumulated experiences of fishers also reflect the seasonal patterns of resources accurately [51,52].

According to our findings, individual brown shrimps can be found in various sizes and densities throughout the ACS throughout the year. The size distribution and density of trawling operations, on the other hand, show clear trends. Fishing grounds I and II have the smallest individuals and are the main recruitment areas. This region is suitable for shrimps because of the overlapping of environmental characteristics such as bottom shape, mud substrate, salinity > 35, and depths of 40 to 60 m [6]. Juveniles and sub-adults make up the majority of those caught in their region. Recruits appear to be using this location as a "steppingstone" to more oceanic settings.

These recruits should migrate for the entire year from densely vegetated coastal areas, where post-larvae development occurs, to offshore locations, with larger concentrations in October–November and then in March–April, indicating two possible recruit cohorts. Southern brown shrimp spawn throughout the year in the ACS, according to the literature, with two peaks, the most significant in June and the less intense in September [12]. Recruitment intensifies four months following each reproductive peak, and lifespan is limited, reaching about 18 months at most [53–55].

Medium-sized classes with likely mature or maturing individuals were found in fishing grounds III and IV. Because of the quick growth of recruits [55], these two recruit cohorts from fisheries grounds III and IV are likely to unite between February and April [12]. This area is also extremely productive, with a larger catch per unit effort [6]. Up to September, mature and reproductive individuals continue to migrate northwest with the main current, arriving at fishing fields V and VI, where shrimps have big size means and smaller catches, perhaps due to a lower abundance of old adults already affected by natural and fishing mortality [21]. Due to the low abundance at that time, most of the fleet returned to fishing grounds I and II in search of better yields, where high-density recruits are starting to arrive.

These findings suggest that *P. subtilis* migrates in the same patterns as other Penaeidae shrimps over the world [3,53]. This study also implies that the recruits migrate north-easterly, following the line of sand and mud river substrate, rather than away to deeper water on the perpendicular to the coast. Young and sub-adult shrimp migration movements offshore correspond to the months of the rainy season, as it does in other shrimp populations [56,57]. The specific ecological mechanism of migration stimulus is unknown. However, it is likely linked to abiotic stimuli including rainfall, runoff, and salinity changes [58,59]. When rainfall is severely low, shrimps have been found to remain in the estuaries without migrating or maturing [60]. Using runoff energy during the rainy season could be a technique to improve individual dispersal on the sea, as well as increase the velocity to reach deeper waters, promote foraging, and avoid predators [61].

The climate and oceanographic features of the region are also linked to the movement pattern's specific characteristics. Juveniles are found in estuaries and shallow coastal waters along the Amazonian coast during the rainy months, primarily in the first semester of the year, where they find a very productive environment with a considerable amount of dissolved and particulate organic material [12,62]. The small-scale artisanal fleet reported the greatest significant harvests of juvenile shrimp between May and July [12]. The second semester's dry period shows a decrease in rainfall. The Amazon River's discharge is at its lowest, and the freshwater plume recedes, increasing salinity in the coastal region, while the North Brazil Current strengthens and flows northwest [35,37,43,63]. These combined environmental conditions may aid shrimp migration to fishery sites near the Amazon and Para River mouths, as well as further northwest.

It might also be stated that the ACS's industrial trawl shrimp fleet's dynamic is more closely linked to shrimp biological traits, with the species being followed in different areas and depths during different stages of their life cycle in order to optimise trip yield [53].

Despite acting on two-yearly cohorts, the fishing fleet is more productive over the offspring from the main reproductive peak in June, catching the young shrimp in shallow seas in October and following them until September in deeper water. Some trawlers return to the recruitment zones in March–April to target the recruits who correspond to the October offspring.

### 4.2. Management Measures and No-Take Zones

Currently, there are certain regulatory measures in place for the industrial trawl shrimp fishery, which include limiting the fishing effort to 101 vessels and a closed season at the end of the year, usually from December to February. Trawling is also forbidden in inshore waters that are fewer than 10 miles from the coast [25,26]. A no-take fishery zone was also proposed by the environment ministry [30], but this proposal was never implemented. Fisheries continue to exist throughout the ACS.

The proposed no-take zone and the current closure season are designed to prevent small individuals from being caught during recruitment. Several authors [64–68] have demonstrated a positive spawning stock–recruitment relationship for shrimps and prawns in several parts of the world. If this relationship is accepted, the most appropriate management measure is to protect the spawning stock, and controlling fishing efforts on the adult population is more important for this than protecting recruits [68].

Shrimps have a classic "r-strategy" life-cycle, with rapid growth and high rates of natural mortality. Fishing efforts do not affect a higher density of adults, because they have an early maturity and continuum breeding with seasonal peaks and recruitment cycles, producing a considerable number of offspring, which can replace deaths. At the same time, the recruitment processes are highly variable, and usually suffer high oscillations depending on environmental factors, such as temperature, oceanographic conditions, winds, salinity, and rainfall, and harvesting as a perturbation source operating combined [54,66,69–74] may affect the water productivity, and be a threat to fisheries resources in ACS [75]. It is a consensus that the joint environmental factors may lead to recruitment failures and affect fisheries, including a drastic decline in biomass, and decreasing catch abundance. Model's predictions on the rainfall patterns in the Eastern Amazon indicated a future with much less rainfall than today [76]. Shrimp recruitment can be threatened in this scenario, and an excessive number of boats can lead to failures in the yield in years with unfavourable environmental conditions.

At ACS, the trawling fishery mostly affects first-generation offspring, and there is no evidence that fishing is having a negative impact on recruitment. Individuals will die of natural mortality if they are not caught by trawlers; hence, intensive effort is normally acceptable for this type of population [3,12,53,54,71]. To avoid overcapitalisation and assure a low effort level after years of recruitment fails, "r-strategist" populations, such as brown shrimp, should be managed by effort management [14,68,77,78]. However, anthropogenic impacts such as climate change, changes in luminosity patterns, rising sea temperatures, and mangrove deforestation may have a greater impact on the shrimp population and fisheries [74,79–81]. Therefore, ecosystem-based management measures should be far more effective than any other control techniques [82].

It is worth noting that in this scenario, no steps are taken to prevent overfishing [53]. According to current research, the ACS brown shrimp stock is in good condition, with sustainable fishing levels [14]. As a result, the best recommendation is to maintain the closure season and trawl licence limits in place. These approaches help to reduce total effort, prevent years of low recruitment and overcapitalisation of the fleet, and improve the logistic and organisational capabilities of industry companies.

Traditional fishery-based management techniques are insufficient to ensure suitable levels of sustainability for this fishery [24], so the imposition of many ecosystem-based management strategies should be explored promptly. They should be considerably more efficient in promoting sustainable fisheries. Ecosystem features, environmental considera-

tions, technological progress, socioeconomic and bioeconomic approaches, as well as stock assessment approaches, are all important parts of fisheries management.

Fishing ground I includes the planned no-take zone. However, it was found that the fleet does not fully utilise this zone. Our data also suggest that the overall shrimp size does not differ statistically across fishing grounds I and II. If the goal of the no-take zone is to protect newly recruited shrimp, and the government insists on enforcing an exclusion zone, the borders should be expanded, from $00°20'$ N to $01°25'$ N and from $47°00'$ W to $48°11'$ W, to be suitable for protecting most newly recruited juveniles of shrimps. The current proposed area should be shifted to the northwest to include fishing grounds I and II (Figure 6). This new area is 16.161 km$^2$, about 1.7 times bigger than the proposed initiative.

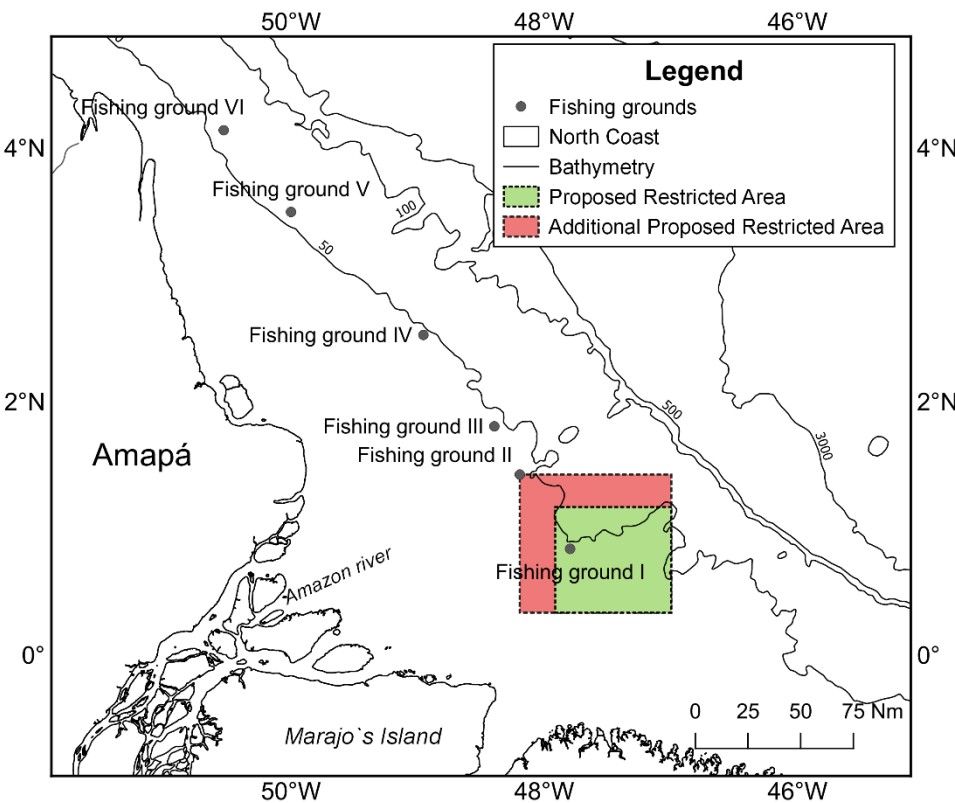

**Figure 6.** The Amazon Continental Shelf, Northern Brazil, with the main fishing grounds and highlighting the original and new enlarged proposed no-take area.

Although there is limited evidence that an exclusion zone benefits the shrimp population, it is crucial to note that this no-take zone can help reduce the quantity of bycatch caught in the trawls of this fishery. Shrimp trawl fisheries produce more than one-third of all bycatch in the world [83]. A no-take zone in front of the Amazon estuary could provide a possibility to preserve some of the rich Amazonian bottom and demersal fauna from trawler activities. However, an economic analysis should be conducted to see whether the economic losses caused as a result of the trawl ban can be compensated for the catch of larger individuals in locations further from the coast. Economic variables can have an impact on the implementation of control measures, and bycatch reduction devices may be more effective in this regard. At the same time, for a poor country such as Brazil with few public policies for the conservation of marine resources, controlling such exclusion zones far from the coast can be a costly and logistically difficult undertaking.

**Author Contributions:** V.J.I. and U.I.P. conceived the original idea, U.I.P. and A.S.M.-F. performed the analyses, U.I.P. wrote the original draft, A.S.M.-F., B.B. and V.J.I. provided critical feedback, review, and editing to the final manuscript; V.J.I. and B.B. supervised the project. All authors have read and agreed to the published version of the manuscript.

**Funding:** This research was funded by the project "Sustainable Management of Bycatch in Latin America and Caribbean Trawl Fisheries–REBYC II," under development by FAO since 2015, and the project "Multidisciplinary cooperative network to support the management of shrimp stocks in the North and Northeast Brazil with an ecosystem focus" Process 445766/2015-8 MCTI/MPA/CNPq. The APC was funded by "PAPIQ-UFPA".

**Acknowledgments:** The authors are grateful to the Centro de Pesquisa e Gestão de Recursos Pesqueiros do Litoral Norte-CEPNOR from Instituto de Conservação da Biodiversidade–ICMBio for providing data. These data were fundamental for the present study. Ualerson I. Peixoto thanks the Conselho Nacional de Desenvolvimento Científico e Tecnologico–CNPq for providing graduate stipends.

**Conflicts of Interest:** The authors declare no conflict of interest.

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
