# Peer review of "Trawl Fishing Fleet Operations Used to Illustrate the Life Cycle of the Southern Brown Shrimp: Insights to Management and Sustainable Fisheries"

_fishes, doi:10.3390/fishes7030141_

Round 1

Reviewer 1 Report

I enjoyed reading this manuscript and learning about this species and fishery. The manuscript is well written, but I have made several minor suggestions throughout the attached PDF (particularly in the introduction).  

The results and discussion are clear.

There are a few statements in the methods where more details could be helpful.  I likely would not have used the statistical methods employed by the authors (i.e., a PERMANOVA with depth classes, etc.). I would have attempted a spatial GAM, with length and depths as continuous variables and (if possible) month as a cyclic variable. However, the methods employed by the authors meet the objectives of the manuscript, so I do not suggest that they be revised.  

Author Response

Response to Reviewer 1 Comments

Dear Review 1

We are pleased with all the considerations that you made in my original article entitled “Trawl fishing fleet operations used to illustrate the life cycle of the southern brown shrimp: Insights to management and sustainable fisheries”.

We appreciate all the contributions of the reviewer to our work. The reviewer 1 made some main general comments and some specific comments.

We accepted all the commentaries and suggestions that you made on the attached PDF. The correction was made directly in the body of the work.

About the general comments;

Point 1: The review 1 asked if the length were skewed, and why the mean was the appropriate metric rather than the median.

Comment 1: The data has not been distorted. In this case, the mean was the best metric because we have a normal equivalence in the size distributions.

Point 2: “I would have attempted a spatial GAM, with length and depths as continuous variables and (if possible) month as a cyclic variable”

Comment 2: We are grateful for reviewer 1's suggestion on new methodologies to be used. We believe that the methodology used in the work met the proposed objectives. But we will certainly take this suggestion into consideration in future works.

Specific comments:

The commas and decimal points were reviewed.

Line 63 – 65: we insert the references.

Line 88-89: excluded.

Line 142: paragraph relocated to the sentence below.

Line 155: we insert where we get the shapes.

Reviewer 2 Report

The authors efficiently described the use of fishing grounds of the Brown shrimp, an important resource for the Countries of Eastern South America, and the relationship existing between one of the main traits of the species (length) and other environmental and anthropogenic variables. I wander why the authors chose the total length and not other measures (i.e., the ratio between the carapace length and weight). I really appreciated the use of the LEK guaranteed by fishermen in the experimental design. I am not an expert of stock assessment (nor shrimp species) and I wonder if the authors considered the sex ratio as a driver of migration or spatial segregation. I really appreciated the discussion highlighting the need for more restrictive measures for the responsible use of this resource and I wonder if genetic approaches have been attempted on this species to verify the connectivity among fishing grounds and if the overcapitalisations/overexploitation is somehow eroding the genetic diversity of the Brown shrimp. I wish this aspect to be considered in the Discussion section.

Very minor comment

Line 223: "P. subtilis" in italics

Author Response

Response to Reviewer 2 Comments

Dear Review 2

We are pleased with all the considerations that you made in my original article entitled “Trawl fishing fleet operations used to illustrate the life cycle of the southern brown shrimp: Insights to management and sustainable fisheries”.

We appreciate all the contributions of the reviewer to our work. The reviewer 2 made some main general comments.

About the general comments;

Point 1: The review 2 asked why the authors chose the total length and not other measures (i.e., the ratio between the carapace length and weight).

Comment 1: The monitoring of shrimp on the Amazon continental shelf is carried out aboard commercial vessels of the industrial shrimp fishing fleet. Because of this, biological samples were taken during fishing operations and due to the logistician on the boat, only the total length was possible at this time. Since shrimp have a high economic value, the companies also did not provide a sample of shrimp to be taken to the laboratory.

Point 2: “I really appreciated the use of the LEK guaranteed by fishermen in the experimental design.”

Comment 2: The use of LEK can help to better understand the spatiotemporal distribution patterns of shrimp sizes on the Amazon continental shelf. This theme is being developed within the scope of another work, which also includes artisanal shrimp fishing in the mangroves of the Amazon.

Point 3: “I am not an expert of stock assessment (nor shrimp species) and I wonder if the authors considered the sex ratio as a driver of migration or spatial segregation”

Comment 2: Due to the limitations of collections exposed in answer 1, the sex of the individuals measured were not observed.

Point 3: “I really appreciated the discussion highlighting the need for more restrictive measures for the responsible use of this resource and I wonder if genetic approaches have been attempted on this species to verify the connectivity among fishing grounds and if the overcapitalisations/overexploitation is somehow eroding the genetic diversity of the Brown shrimp.”

Comment 3: New works that are testing different models of spatial and/or temporal management measures are being developed. These works can give us a better basis for proposing more precise management measures. For now, we do not have data that can answer genetic questions.

Specific comments:

Line 223: The name was put in italic.
